# Effect of Thermal Inactivation on Antioxidant, Anti-Inflammatory Activities and Chemical Profile of Postbiotics

**DOI:** 10.3390/foods12193579

**Published:** 2023-09-26

**Authors:** Zhe Sun, Zhi Zhao, Bing Fang, Weilian Hung, Haina Gao, Wen Zhao, Hanglian Lan, Mingkun Liu, Liang Zhao, Ming Zhang

**Affiliations:** 1School of Food and Health, Beijing Technology and Business University, Beijing 100048, China; 2Key Laboratory of Precision Nutrition and Food Quality, Department of Nutrition and Health, China Agricultural University, Beijing 100193, China; 3Inner Mongolia Dairy Technology Research Institute Co., Ltd., Hohhot 010110, China; 4National Center of Technology Innovation for Dairy, Hohhot 010110, China; 5School of Food and Biological Engineering, Hefei University of Technology, Hefei 230009, China; 6Key Laboratory of Functional Dairy, College of Food Science and Nutritional Engineering, China Agricultural University, Beijing 100083, China

**Keywords:** postbiotics, *Lacticaseibacillus paracasei*, *Bifidobacterium lactis*, thermal inactivation, antioxidant, anti-inflammatory, chemical profile

## Abstract

Inactivation is a crucial step in the production of postbiotics, with thermal inactivation being the prevailing method employed. Nevertheless, the impact of thermal treatment on bioactivity and chemical composition remains unexplored. The objective of this study was to assess the influence of heating temperature on the antioxidant, anti-inflammatory properties and the chemical composition of ET-22 and BL-99 postbiotics. The findings revealed that subjecting ET-22 and BL-99 to thermal treatment ranging from 70 °C to 121 °C for a duration of 10 min effectively deactivated them, leading to the disruption of cellular structure and release of intracellular contents. The antioxidant and anti-inflammatory activity of ET-22 and BL-99 postbiotics remained unaffected by mild heating temperatures (below 100 °C). However, excessive heating at 121 °C diminished the antioxidant activity of the postbiotic. To further investigate the impact of thermal treatments on chemical composition, non-targeted metabolomics was conducted to analyze the cell-free supernatants derived from ET-22 and BL-99. The results revealed that compared to mild inactivation at temperatures below 100 °C, the excessive temperature of 121 °C significantly altered the chemical profile of the postbiotic. Several bioactive components with antioxidant and anti-inflammatory properties, including zomepirac, flumethasone, 6-hydroxyhexanoic acid, and phenyllactic acid, exhibited a significant reduction in their levels following exposure to a temperature of 121 °C. This decline in their abundance may be associated with a corresponding decrease in their antioxidant and anti-inflammatory activities. The cumulative evidence gathered strongly indicates that heating temperatures exert a discernible influence on the properties of postbiotics, whereby excessive heating leads to the degradation of heat-sensitive active constituents and subsequent diminishment of their biological efficacy.

## 1. Introduction

Recent studies on probiotics proved that the viability of probiotics is not strictly required for their health benefits [1]. Some inactivated probiotics also had the same or similar probiotic effects. As new evidence arises continuously, the concept of postbiotic was conceived. In 2019, the International Scientific Association of Probiotics and Prebiotics released consensus statement on the definition postbiotics [2]. Postbiotics were defined as a preparation of inanimate microorganisms and/or their components that confers a health benefit on the host [2].

Since the release of consensus, postbiotics has rapidly gained immense attention of academia and industry, although the rigor of concept has not been unanimously recognized by all academicians [3]. Among them, some unique advantages of postbiotics have been repeatedly mentioned, such as biological security, storage stability and better recognitive accessibility with Pattern Recognition Receptors [4]. Cuevas-González et al. (2020) provided a comprehensive overview of the current understanding of the functional and biological activities of postbiotics, including antimicrobial, antioxidant, and immunomodulatory activities, which can be exerted through direct or indirect mechanisms [5]. Many postbiotics consist of non-living strains that established probiotic taxa such as *Bifidobacterium* and *Lactobacillus*. Moreover, there are also specific strains, such as *Akkermansia muciniphila* and *Faecalibacterium prausnitzii*, that meet the criteria for postbiotics and have been referred to as the “next-generation probiotics” in previous research [6,7,8]. 

Microbial inactivation technology is a necessary step in the preparation postbiotics [9]. Therefore, the selection of inactivation technology and the morphology and bioactivity changes of bacteria under different inactivation processes are the primary concerns. The present study conducted a comparative analysis of the effects of various inactivation methods on the integrity and morphology of probiotics cells [10]. Specifically, the impact of ohmic heating on the viability and morphology of probiotic cells was evaluated [11]. Pimentel et al. (2023) provided a summary of the inactivation processes and parameters utilized in recent studies, ultimately concluding that thermal treatment is the prevailing method [12]. However, the postbiotics are a complex mixture of metabolic products secreted in supernatants, including lipids, nucleotides, organic acids, and peptides [13]. It is important to note that the temperature and duration of heating can influence the composition and biological activity of postbiotics. However, thus far, there has been limited investigation into the effects of the inactivation process on biological activity.

Our previous studies have reported that *Lacticaseibacillus paracasei* ET-22 (ET-22) [13,14] and *Bifidobacterium lactis* BL-99 (BL-99) [15] exhibited the benefit effects of modulating microflora, inhibiting pathogenic biofilm and relieving inflammation. The objective of the present study was to investigate the impact of thermal treatment on antioxidant, anti-inflammatory activities of ET-22 and BL-99 postbiotics. Furthermore, non-targeted metabolomics, as a useful tool for microbial metabolites analysis [16], was performed to investigate the differences in chemical profile under different heat treatment conditions. It is expected that these findings will provide a theoretical basis for the processing of ET-22 and BL-99 postbiotics.

## 2. Materials and Methods

### 2.1. Bacterial Strains

*Lacticaseibacillus paracasei* ET-22 (ET-22) and *Bifidobacterium lactis* BL-99 (BL-99) were provided by Yili Innovation Center, Inner Mongolia Yili Industrial Group Co., Ltd, Hohhot, China. ET-22 and BL-99 were cultured in MRS Broth at 37 °C overnight before harvest. BL-99 was incubated in an anaerobic environment, which was kept by Anaero sachets (Oxoid Ltd., West Heidelberg, Australia).

### 2.2. Preparation of Postbiotic

The overnight cultures of ET-22 and BL-99 were harvested by centrifugation (4500× *g*, 10 min), washed three times and resuspended in PBS to a concentration of 1 × 10^10^ CFU/mL. As for postbiotic preparation, the heat-killed bacteria were inactivated heating at 70 °C, 80 °C, 90 °C, 100 °C, and 121 °C for 10 min. The number of residual live cells was counted by MRS plate. The cell-free supernatant was collected as another postbiotic after centrifugation at 10,000× *g* for 10 min at 4 °C, and filtered through 0.22 μm sterile aqueous filter membrane.

### 2.3. Morphological Changes of Postbiotic by Scanning Electron Microscopy (SEM)

The morphological examination of live and heat-killed bacteria was observed by SEM. The live or inactivated cells were fixed in 2.5% glutaraldehyde fixative at 4 °C, then dehydrated in different gradients (50%, 60%, 70%, 80%, 90%, 100%) of ethanol. On the stubs, slides were mounted and sputtered with gold after critical point drying. The morphology changes were then examined by with SU8020 SEM (HITACHI, Hitachi, Japan).

### 2.4. Antioxidant Activity Assay of Postbiotic

#### 2.4.1. 2,2-Diphenyl-1-Picryl-Hydrazyl (DPPH) Free Radical Scavenging

The DPPH radical scavenging assay was measured using the previous method with minor modification [17]. 500 μL of each sample and ethanolic DPPH were vigorously mixed, followed by incubation at 25 °C in the dark for 30 min. In the control group, samples were replaced with PBS (pH 7.4) in equal volume, while the DPPH radical solution was replaced with an equal amount of PBS (pH 7.4) in the blank group. The absorbance of the solution was measured at 517 nm after centrifugation at 2000× *g* for 10 min.
$$\mathrm{Scavenging}\;\mathrm{ability}\;(\%)=[1-\frac{A_{\mathrm{sample}}-A_{\mathrm{blank}}}{A_{\mathrm{control}}}]\times 100$$

#### 2.4.2. Hydroxyl Radical Scavenging

The hydroxyl radical scavenging test was performed as described by Chang et al. [18]. Each 1 mL sample was mixed with 1,10-phenanthroline, PBS (pH 7.4), and FeSO_4_ prior mixed with H_2_O_2_. After incubating at 37 °C for 90 min, the resulting solution was analyzed at 536 nm, and the hydroxyl radical scavenging activity was calculated as follows:$$\mathrm{Scavenging}\;\mathrm{ability}\;(\%)=(\frac{A_{\mathrm{sample}}-A_{\mathrm{blank}}}{A_{\mathrm{control}}-A_{\mathrm{blank}}})\times 100$$

### 2.5. Anti-Inflammatory Activity Assay of Postbiotic

RAW 264.7 macrophages were cultured in DMEM containing 10% FBS (BI, Israel) and 1% penicillin/streptomycin (Gibco, Brisbane, Australia). To determine the anti-inflammatory activity, the RAW264.7 macrophages (1 × 10^6^ cells/mL) were seeded in a 12-well plate, and then stimulated with LPS (1 ug/mL, Sigma-Aldrich, Shanghai, China) for 24 h. Subsequently, different postbiotics incubated with cells for 24 h. The concentrations of postbiotics, including heat-killed cells and cell-free supernatant, were calculated as the number of viable cells (5 × 10^8^ CFU/mL) before inactivation. 

To investigate the effect of different postbiotics on inflammatory factors, the reverse transcription method was used to obtain cDNA from total RNA extracted with TRIzol reagent (Invitrogen, Waltham, MA, USA). Quantitative real-time PCR reactions were performed with SYBR Green PCR Master Mix (TaKaRa, Shiga, Japan). Specific primer sequences were as follows: *TNF-α*, forward 5-CTGAACTTCGGGGTGATCGG-3, reverse 5-GGCTTGTCACTCGAATTTTGAGA-3; *GAPDH*, forward 5-AAGCCCATCACCATCTTCCA-3, reverse 5-CACCAGTAGACTCCACGACA-3. We determined the relative expression of each target gene by 2^−ΔΔCt^ method. All quantifications were normalized to the GADPH gene.

### 2.6. Chemical Profiles Assay by Non-Targeted Metabonomics

To compare the metabolites of the cell-free supernatant, a non-targeted metabonomic was performed. The sample preparation referred to our previous study [13]. LC-MS/MS determination was performed by Thermo UHPLC-Q Exactive HF-X Mass Spectrometer. After separation by an HSS T3 column, the samples were subjected to MS/MS detection for compound identification. Q Exactive HF-X mass spectrometer was operated as follows: −3500 V in negative mode and 3500 V in positive mode, capillary temperature at 325 °C, heater temperature at 425 °C, MS resolution at 60,000, MS/MS resolution at 75,000. Mass range was set to 70–1050. After determination, Metlin (https://metlin.scripps.edu/, accessed on 23 May 2023) and HMDB (http://www.hmdb.ca/, accessed on 3 June 2023) were used to identify the metabolites. The data were analyzed by Majorbio cloud platform (https://cloud.majorbio.com, accessed on 4 June 2023). Detailed information about the data filtered, data normalized, and data analyzed can be found in our previous study [13].

### 2.7. Statistical Analysis

Data are presented as the mean ± standard deviation, and all experiments were performed at least in triplicate. In order to compare between groups, a two-tailed Student’s *t*-test and an analysis of variance (ANOVA) with Dunnett’s multiple comparison test were used. Data analyses were conducted using GraphPad Prism.

## 3. Results

### 3.1. Effects of Inactivation Treatments on Cell Morphology by SEM

To confirm the inactivation effect, residual viable cells were counted by the traditional plating method. The results showed that thermal treatment ranging from 70 °C to 121 °C for a duration of 10 min effectively deactivated ET-22 and BL-99.

SEM imaging was then used to visualize the morphological alterations of ET-22 and BL-99 after thermal treatment. As shown in Figure 1, the live ET-22 cells appeared smooth and regular with intact membranes. While the morphology of bacterial cells treated at 70 °C showed obviously changed, as evidenced by the appearances of cell shrinkage and white clustered particles. As the temperature continued to rise, cell shrinkage and breakage of the bacterial cells continuously presented, accompanied by white clustered particles and leakage of intracellular content (Figure 1).

Regarding BL-99, the typical morphology of Bifidobacterium was readily observable, with a smooth and intact surface (Figure 2). Notably, the application of heat treatment at 70 °C seemed to have some impact compared to ET-22 cells, resulting in increased surface roughness and white clustered particles on the basis of cell death. As the temperature increased further, the release of intracellular content and cell debris became more pronounced in comparison to ET-22 cells at the same temperature (Figure 2). These findings suggest that the thermal treatment methods employed in this study effectively deactivate probiotics, and with the increase in thermal strength, the degree of cell shrinkage and breakage increased. ET-22 has greater resistance to thermal conditions than BL-99.

### 3.2. Effects of Inactivation Treatments on Antioxidant Activity

#### 3.2.1. DPPH Radical Scavenging Capacity

DPPH scavenging ability is a common evaluation metrics to assess the antioxidant activity of bioactive components. As shown in Figure 3A, ET-22 live cells exhibited a strong DPPH free radical scavenging ability, and radical scavenging effects reached 92.96%. Compared with live bacteria, no significant reduction in scavenging ability was observed when ET-22 was treated at 70 °C or 80 °C. With an increase in temperature above 90 °C, activity began to decrease significantly (*p* < 0.05). Specifically, effective radical scavenging fell below 65.21% with 121 °C heating treatment. Regarding BL-99, the DPPH scavenging abilities of both live and heat-killed cells was significantly weaker than ET-22. The DPPH scavenging capacity of the live BL-99 cells was 36.02%. With the increase in heating temperature, the scavenging abilities did not change significantly compared with live cells. However, similar to ET-22, when the inactivation temperature increased to 121 °C, scavenging capacity significantly dropped to 23.45% (Figure 3B).

Cell-free supernatants play a crucial role in postbiotic composition. In this study, we conducted a comparison between the DPPH radical scavenging capacity of cell-free supernatants and whole heat-killed cells. The results, as depicted in Figure 3C, demonstrated that regardless of whether the cells were subjected to 80 °C or 121 °C treatments, the DPPH scavenging abilities of cell-free supernatants were significantly lower than those of heat-killed cells at equivalent doses (*p* < 0.05). Furthermore, when compared to BL-99 heat-killed cells, the DPPH scavenging abilities of cell-free supernatants exhibited varying degrees of decline (Figure 3D). These findings suggested that cell-free supernatants of ET-22 and BL-99 contributed to a portion of the antioxidant activity.

#### 3.2.2. Hydroxyl Radical Scavenging Capacity

The hydroxyl free radical scavenging ability is another indicator to assess free radicals scavenging abilities for active ingredients. As shown in Figure 4A, scavenging ability of ET-22 live cells was 29.31%. Interestingly, ET-22 heat-killed cells had a significantly higher scavenging ability at different temperatures than live cells (*p* < 0.01). As the temperature increased, the scavenging ability of heat-killed cells showed small variations, but higher than the ability of live cells (Figure 4A). As for BL-99, the scavenging ability of live cells was significantly higher than ET-22 (Figure 4B). Moreover, scavenging capacities heat-killed cells remained unaffected by mild heating temperatures (below 100 °C). However, excessive heating at 121 °C significantly diminished the scavenging capacity of the postbiotic (Figure 4B).

We also conducted a comparison between the hydroxyl free radical scavenging capacities of cell-free supernatants and whole heat-killed cells. As shown in Figure 4C, either after 80 °C or 121 °C treatments, ET-22 heat-killed cells showed better scavenging ability compared to cell-free supernatants. Similarly, cell-free supernatants of BL-99 had weaker scavenging ability than heat-killed cells (Figure 4D). 

These findings indicated that inactivation treatments significantly affected the antioxidant activity of postbiotics, mild heating temperature had no significant effect on antioxidant activity, and even contributes to the release of some antioxidant components. However, the excessive heating temperature (121 °C) would significantly reduce the antioxidant activity of postbiotic. As an important component of postbiotic, cell-free supernatants provided part of the antioxidant capacity for postbiotics.

### 3.3. Effects of Inactivation Treatments on Anti-Inflammatory Activity

TNF-α production is associated with acute and chronic inflammatory diseases. To further study the anti-inflammatory activity of different postbiotics, the expression of pro-inflammatory factor TNF-α was detected by RT-qPCR. As shown in Figure 5A, LPS induces an inflammatory response in macrophages cells, the expression TNF-α was significantly elevated, while heat-killed ET-22 effectively suppressed this elevation (*p* < 0.05). ET-22 samples that were treated at 80 °C and 100 °C demonstrated greater effects than ET-22 samples treated at 121 °C (*p* < 0.05), which implied excessive heating temperature might destroy some active ingredients. In addition, compared with ET-22, heat-killed BL-99 exhibited better thermal stability in anti-inflammatory activity. The anti-inflammatory effect of postbiotic obtained at different temperatures was basically consistent, and all of them were significantly lower than those in the LPS treatment group (*p* < 0.05). 

In this study, a comparison was conducted to assess the anti-inflammatory capacity of cell-free supernatants and whole heat-killed cells. The results depicted in Figure 5C,D indicated that there were no statistically significant differences in TNF-α expression between heat-killed cells and cell-free supernatants obtained at 80 °C (*p* > 0.05). However, the inhibitory effect of cell-free supernatant obtained at 121 °C was found to be more significant than that of heat-killed bacteria (*p* < 0.01). Furthermore, the cell-free supernatants of BL-99 exhibited a significant inhibition of TNF-α expression (*p* < 0.05), although the inhibitory effect was inferior to that of heat-killed bacteria under the same conditions. 

The above results suggested that the effect of inactivation treatments on anti-inflammatory activity may be strain-specific. There was still strong anti-inflammatory activity in postbiotics treated at mild temperatures. This was largely due to the cell-free supernatants of the postbiotics. 

### 3.4. Non-Targeted Metabolomics Analysis of the Postbiotic Chemical Profiles 

Metabolomics has been involved in the identification of metabolites in biological systems, which could help to investigate postbiotic metabolites [16]. In the current study, non-targeted metabolomics was performed to analyze the chemical composition of the cell-free supernatants of derived from ET-22 and BL-99 with different thermal treatments. A total of 2620 substances was identified by the HMDB and Metlin databases, including amino acids, organic acids, amines, nucleotides, lipids, peptides, and so on. 

PLS-DA analysis was performed to investigate the distinctions of different cell-free supernatants. As shown in Figure 6A, two cell-free supernatants separated totally, and the PLS-DA axis 1 and axis 2 explained 63.2% and 20.99% of the total variation. More importantly, the postbiotic obtained from same strains showed totally different aggregation characteristics on the PLS-DA plot, simply due to the difference in heating temperature. The chemical composition of ET-22 or BL-99 cell-free supernatants heated at 121 °C was significantly different from that of 80 °C and 100 °C (Figure 6A). Correlation analysis further verified the PLS-DA results (Figure 6B). Cell-free supernatants obtained at 80 °C and 100 °C had similar chemical compositions, while the similarity decreased significantly after the temperature was further increased. The trend of BL-99 was almost consistent with ET-22 (Figure 6B). These results demonstrated that excessive heating temperature (121 °C) would change the chemical profile of a postbiotic, which is perhaps related to declination in antioxidant and anti-inflammatory activity.

### 3.5. Differential Metabolites Analysis of Different Cell-Free Supernatants 

Heatmap was performed to depict the metabolites composition of cell-free supernatants obtained at different heating temperatures. As shown in Figure 7A, cell-free supernatants obtained at 121 °C remarkably changed, and 12 compounds with high levels at 80 °C and 100 °C almost disappeared, such as zomepirac, ADP ribose, flumethasone, adenosine diphosphate ribose, citramalic acid, 9,10-epoxy-18-hydroxy-octadecanoic acid, 6-hydroxyhexanoic acid, phenyllactic acid, 2-amino-3-[[(2R)-2-carboxy-2-(diacetylamino)ethyl]disulfanyl]propanoic acid, O-succinyl-L-homoserine, D-2-hydroxyglutaric acid, and TriHOME. Meanwhile, high-temperature treatment also produced a series of new compounds, including 3-(4-hydroxyphenyl) lactate, xanthine, pseudouridine 5-phosphate, 5-thymidylic Acid, D-(+)-malic acid, adenine, D-manno-2-heptulose, 3-aminopentanedioic acid, tyrosine methylester, glucose propionate. The cell-free supernatants treated at 80 °C and 100 °C have a high similarity in the composition of the main compound, and there are only differences in the abundance of 8 compounds, including L-4-Hydroxyglutamate semialdehyde, 5-CMP, Penicilloic acid, 3-Adenylic Acid, Adenosine 2-phosphate, L-beta-aspartyl-L-leucine, N-Acetyl-L-glutamate 5-semialdehyde, and Malic acid. 

As a heat-sensitive strain, the cell-free supernatants of BL-99 also changed significantly with the increase in inactivation temperature (Figure 7B). Particularly when the temperature reached 121 °C, 13 compounds that were highly enriched in the low-temperature samples disappeared, such as D-2-hydroxyglutaric acid, 3-hydroxyamobarbital, 8-lsobutanoylneosolaniol, shanzhiside methyl ester, morroniside, 2-isopropylmalic acid, 3-(4-hydroxyphenyl) lactate, L-beta-aspartyl-L-leucine, 6-hydroxyhexanoic acid, 4-hydroxycyclohexylcarboxylic acid, phenyllactic acid, antiarrhythmic peptide, and glutarylglycine. Meanwhile, a series of new compounds were generated after heating at 121 °C, including N-acetyl-L-glutamic acid, tyrosine methylester, 3-adenylic acid, pseudouridine 5-phosphate, D-arabitol, Beta-L-fucose 1-phosphate, L-4-hydroxyglutamate semialdehyde, D-(+)-malic acid, 9,10-epoxy-18-hydroxy-octadecanoic acid, 5-thvmidvlic acid, and 3-aminopentanedioic acid. The above results implied that chemical composition of postbiotic was very diverse and variable, even if it is derived from the same probiotic. This diversity and variability may be further amplified by differences in inactivation temperatures and strain origins.

## 4. Discussion

Postbiotic is a constantly evolving concept [1], and the current studies about postbiotics have been restricted to the benefit effect. Inactivation technology is a necessary step in the preparation postbiotics, but the effects of inactivation process on biological activity have hardly been involved. Here, the current study demonstrated that mild heating temperature (below 100 °C) had no significant effect on antioxidant and anti-inflammatory activity of ET-22 and BL-99 postbiotics, but the excessive heating temperature (121 °C) would significantly reduce the antioxidant activity of postbiotic. Several bioactive components with antioxidant and anti-inflammatory properties exhibited a significant reduction in their levels following exposure to a temperature of 121 °C. This decline in their abundance may be associated with a corresponding decrease in their antioxidant and anti-inflammatory activities.

Since the composition of postbiotic was complex, the mechanism of its biological activity and the signaling pathway involved have not been fully elucidated [19]. Thermal treatment is the most common process used for microbial inactivation [12]. Studies have found that the postbiotic with thermal treatment has a wide range of biological activities [20,21]. However, which component provide the host with biological activity is the key question. One previous studied found that the peptidoglycan of *Lactobacillus plantarum* CRL1505 improved the innate immune response of malnourished mice, and retained the complete immunomodulatory properties of live bacteria [22]. Kim et al. proved that S-layer protein was the bioactive component of Kefir *Lactobacillus* and improved inflammation and insulin resistance [23]. Khmaladze et al. demonstrated that cell lysates of *Lactobacillus reuteri* DSM 17938 reduced UV-R-induced inflammatory cytokines IL-6 and IL-8 and protected the skin barrier [24]. Wu et al. found that the cell-free supernatant of *Akkermansia muciniphila*, as a postbiotic, has the potential to prevent obesity and improve disorders of glucose metabolism [25]. 

In our study, SEM was used to find that the inactive cells showed obvious shrinkage and breakage, but some cells were still intact. Therefore, heat-killed postbiotic often appear in the form of a mixture of intact inactive cells, cell wall and membrane, and exuded intracellular fraction. The diversities of components of postbiotic mean that compared with live cells, postbiotic could bind to more receptors, activating more signaling pathways. For example, exopolysaccharides bind TLR2/TLR4 specifically, while LTA and active peptides are more likely to bind TLR2 and 6 [26]. Although heat-killed postbiotics do not have the ability of intestinal colonization, the diversification of postbiotics may be able to stimulate multi-level biological responses at the same time. Cross-reactions between biological responses may be more complex and beneficial. 

Health effects associated with postbiotics are widespread, with antioxidant and anti-inflammatory being the most common health benefits. Research has suggested that heat-killed *Lactobacillus brevis* B13-2 (85 °C, 30 min) showed higher ABTS radical scavenging activity and β-carotene oxidation inhibition activity than live bacteria [27]. Both live bacteria and heat-killed *Lactobacillus plantarum* NA (121 °C, 15 min) possessed capacity for scavenging free radicals [28]. Heat-killed *Weissella cibaria* JW15 (90 °C, 30 min) showed anti-inflammatory effects by inhibiting NF-κB activation and reducing the pro-inflammatory function of LPS-induced RAW 264.7 cells [29]. Heat-killed LGG (85 °C, 30 min) reduced the production of LPS-induced pro-inflammatory factors and increase anti-inflammatory factors [30]. As mentioned in a previous review [12], the thermal inactivation conditions of the postbiotics in the above studies were inconsistent, and the effects of thermal treatment against bioactivity and chemical composition have not been reported. The current study proved that mild heating temperature (below 100 °C) had no significant effect on antioxidant and anti-inflammatory activity of ET-22 and BL-99 postbiotics, but the excessive heating temperature (121 °C) would reduce both and antioxidant and anti-inflammatory activities of postbiotic. These findings are expected to provide some guidance for postbiotics preparation.

The cell-free supernatant is an important part of postbiotic, and its importance is also of great concern in the postbiotic. Rocchetti et al. found that the cell-free supernatant of *L. plantarum* also had similar immunomodulatory capacity compared with live bacteria [31]. RAMOS et al. demonstrated that the cell-free supernatant of *Lactiplantibacillus plantarum* 6.2 excluded *Gardnerella vaginalis* from the adhesion site, and has good bactericidal effect like live bacteria [32]. Our study suggested that the antioxidant and anti-inflammatory capacity of the cell-free supernatant was lower than that of whole postbiotics at the same dose, but the biological activity of cell-free supernatants is still not negligible, especially in terms of anti-inflammatory capacity. Other components of the bacteria, such as EPS, LTA, etc., may provide additional anti-inflammatory and antioxidant capabilities [26]. Furthermore, from a biosafety perspective, cell free metabolites secreted by probiotic strains have been proposed as a better and safer strategy [33].

Non-targeted metabolomics is a useful tool for microbial metabolites analysis, especially for small molecule metabolites [16]. Xu et al. analyzed the microbial composition changes during the brick-tea fermentation by non-targeted metabolomics [34]. Subsequently, metabolomics has been widely used for microbial metabolites analysis during multi-species and single-specie fermentation [35,36]. In the current study, we performed non-targeted metabolomics in the analysis of postbiotic meta-systems and identified 2620 substances. Although the metabolomics does not allow for content calculation of the specific substance, we were still able to compare the relative amounts of substances.

The relationship between inactivation process of probiotic and bioactivity is not clear. Song et al. found that *L. rhamnosus* GG and *L. brevis* B13-2 after heat inactivation showed a decreasing trend of DPPH free radical scavenging ability [27]. Xu et al. proved that the anti-inflammatory activity of *Lactiplantibacillus plantarum* NA after heat inactivation at 10^8^ cfu ml^−1^ was decreased and produced more pro-inflammatory factors TNF-α and IL-6 [28]. However, they did not clarify the relationship between changes of metabolites after probiotics inactivation. The results in our study demonstrated that excessive heating temperature (121 °C) would significantly reduce the antioxidant and anti-inflammatory activity of postbiotic. Consistent with these results, metabolomics also proved that the chemical composition of ET-22 after 121 °C treatment was significantly changed. Zomepirac, flumethasone, 6-hydroxyhexanoic acid, and phenyllactic are several microbial metabolites with well-defined anti-inflammatory effects [37,38,39,40], enriched in postbiotics which inactivated below 100 °C, but virtually disappeared after heating at 121 °C. The quenching of the active metabolites may directly contribute to the decrease in anti-inflammatory activity. Meanwhile, the decline of phenyllactic acid, which also confers antioxidant effect [41], also affected the antioxidant capacity of the postbiotic to some extent. Similarly, for BL-99, the decrease in 6-hydroxyhexanoic acid and phenyllactic acid after 121 °C inactivation may have contributed to the decrease in the antioxidant and anti-inflammatory activities. Therefore, the decrease in the content of metabolites with antioxidant and anti-inflammatory activity may lead to a decrease in biological activity of probiotics. However, the current study only proved that changes in the content of specific metabolites had the same trend as biological activities such as antioxidants or anti-inflammatory. More studies are needed to be able to establish the relationship between changes in activity and changes in a specific compound.

## 5. Conclusions

In this study, the effects of thermal temperature on antioxidant, anti-inflammatory activities of ET-22 and BL-99 postbiotics were investigated. The results showed that thermal treatment ranging from 70 °C to 121 °C for a duration of 10 min can effectively inactivate ET-22 and BL-99, resulting in the breakage of the cell structure and leakage of intracellular contents. Mild heating temperature (below 100 °C) had no significant effect on antioxidant and anti-inflammatory activity of ET-22 and BL-99 postbiotics, but the excessive heating temperature (121 °C) remarkably diminished the antioxidant activity of the postbiotic. Non-targeted metabolomics was then performed to analyze the cell-free supernatants of derived from ET-22 and BL-99. Compared with the mild inactivation (below 100 °C), excessive temperature (121 °C) significantly changed the chemical profile of postbiotic. Several bioactive components with antioxidant and anti-inflammatory properties, including zomepirac, flumethasone, 6-hydroxyhexanoic acid, and phenyllactic acid, exhibited a significant reduction in their levels following exposure to a temperature of 121 °C. This decline in their abundance may be associated with a corresponding decrease in their antioxidant and anti-inflammatory activities. The cumulative evidence gathered strongly indicates that heating temperatures exert a discernible influence on the properties of postbiotics, whereby excessive heating leads to the degradation of heat-sensitive active constituents and subsequent diminishment of their biological efficacy.

## Figures and Tables

**Figure 1 foods-12-03579-f001:**
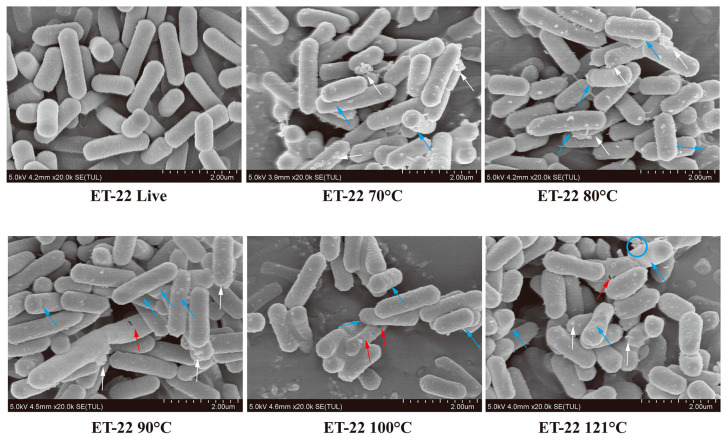
Typical SEM photographs of viable and thermal inactivated *Lacticaseibacillus paracasei* ET-22. Blue arrow: cell shrinkage; Red arrow: breakage of the bacterial cells; white arrow: white clustered particles; circle: intracellular content.

**Figure 2 foods-12-03579-f002:**
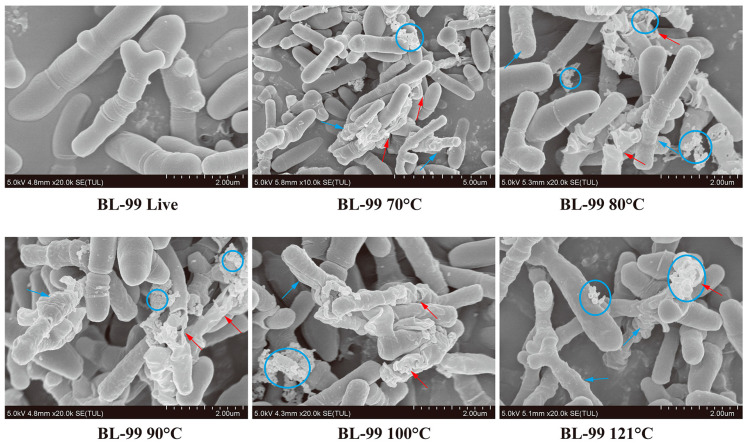
Typical SEM photographs of viable and thermal inactivated Bifidobacterium animalis subsp. lactis. BL-99. Blue arrow: cell shrinkage; Red arrow: breakage of the bacterial cells; circle: leakage of intracellular content.

**Figure 3 foods-12-03579-f003:**
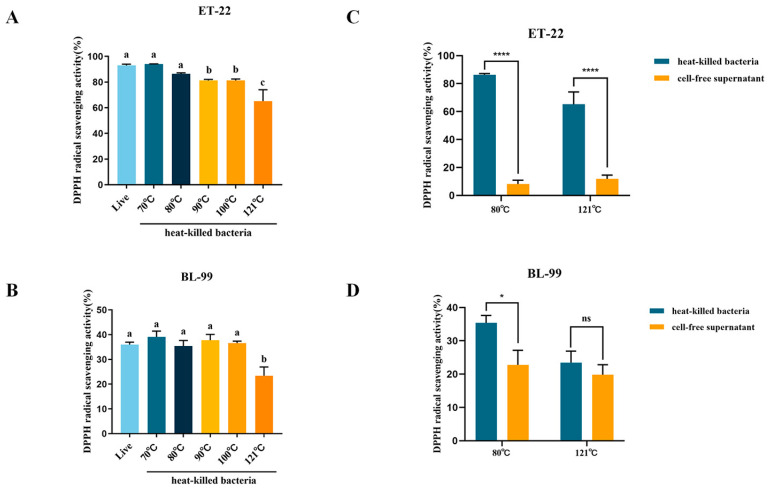
Effects of inactivation treatments on DPPH scavenging abilities. (**A**,**B**) Heat-killed ET-22 and BL-99. (**C**,**D**) Heat-killed bacteria and cell-free supernatants. (a, b, c, means with the different letter are significantly different (*p* < 0.05); * *p* < 0.05; **** *p* < 0.0001).

**Figure 4 foods-12-03579-f004:**
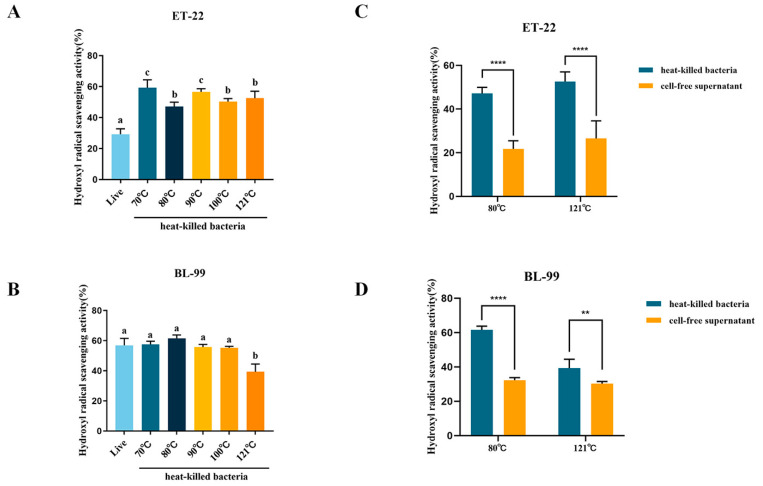
Effects of inactivation treatments on hydroxyl radical scavenging **abilities**. (**A**,**B**) The abilities of heat-killed ET-22 and BL-99; (**C**,**D**) Heat-killed bacteria and cell-free supernatants. (a, b, c, means with the different letter are significantly different (*p* < 0.05); ** *p* < 0.01; **** *p* < 0.0001).

**Figure 5 foods-12-03579-f005:**
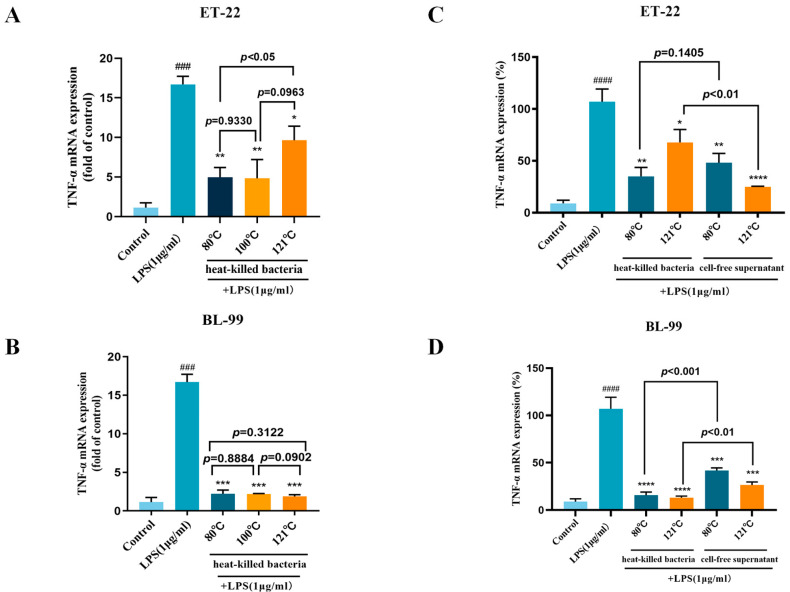
Effects of inactivation treatments on anti-inflammatory activity. (**A**,**B**) The effect of heat-killed ET-22 and BL-99; (**C**,**D**) Heat-killed bacteria and cell-free supernatants. (###, *p* < 0.001, ####, *p* < 0.0001, significant different with control group; * *p* < 0.05; ** *p* < 0.01; *** *p* < 0.01; **** *p* < 0.0001, significant different with LPS group).

**Figure 6 foods-12-03579-f006:**
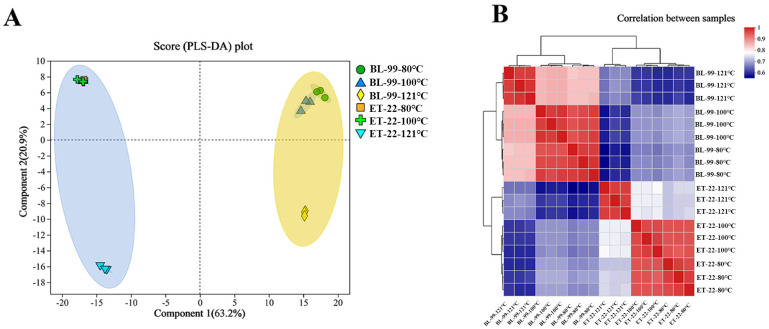
Multivariate analysis of the chemical composition of different thermal inactivated cell-free supernatants. (**A**) PCA score scatter plot composed of chemical composition (PC1 = 63.2%, PC2 = 20.9%); (**B**) Correlation analysis between cell-free supernatants of ET-22 and BL-99.

**Figure 7 foods-12-03579-f007:**
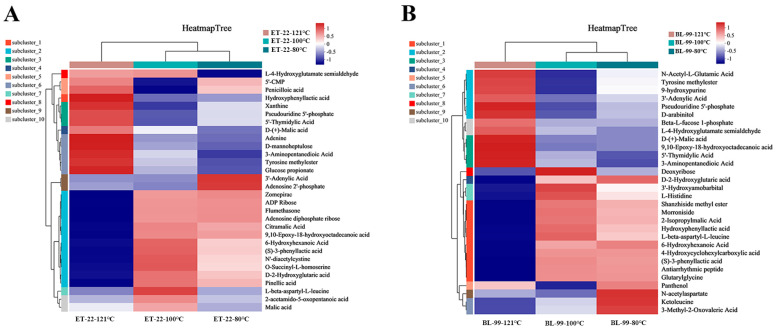
Heatmap cluster analysis of chemical composition of different cell-free supernatants. (**A**) ET-22; (**B**) BL-99.

## Data Availability

The data presented in this study are available herein.

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
