# Peer review of "Effect of Thermal Inactivation on Antioxidant, Anti-Inflammatory Activities and Chemical Profile of Postbiotics"

_foods, 2023, doi:10.3390/foods12193579_

Round 1
Reviewer 1 Report
The work is very interesting and addresses a novel topic.
I recommend improving the description of the methodologies.
I consider it necessary to highlight the importance of the inactivation results and their relationship with the effects observed with respect to antioxidant activity, the same case in anti-inflammatory activity, the relationship between cellular inactivation is not clear, I consider that more studies are needed to be able to conclude that the temperature will decrease the biological effects.
English is correct
Author Response
Response:
Thank you very much for your suggestions. Your suggestions are of vital importance. We agree with you that the relationship between inactivation process of probiotic and bioactivity is not clear. In this study, we only proved that changes in the content of certain actives had the same trend as biological activities such as antioxidants or anti-inflammatory. In our ongoing research, we seek to clearly establish the relationship between changes in activity and changes in a specific compound. Based on your suggestion, we have revised the last paragraph of the discussion to provide a clearer picture of the progress and shortcomings of the current study.

Reviewer 2 Report
The paper is good. I suggest for minor revisions.
Page 1, line 24 (Abstract): “Significantly” means that a statistically reduction occurred?
Page 2, lie 40-41: I would say “Recent studies on probiotics proved that… “
Page 2, line 51: I think that this sentence could be removed. It is clear that researchers are focusing on this topic.
Page 2, line 53: The year of the study is missing. Please, add it.
Page 2, lines 52-56: Specify which properties.
Page 2, line 57: Replace “like” with “such as.”.
Page 2, line 60: The sentence could be removed.
Page 2, L 68: Add the year of the publication.
Page 2, L72: Significantly” means that a statistically reduction occurred? If it’s not a statistical reduction, please use adverbs such as “Relevantly” or “Importantly” or synonymous.
Page 2, L75: Which previous studies? Can you add them as references?
Page 5, LL 171-172: “…more significant impact compared to ….” is not sufficiently clear. I suggest to specify how much the impact was more significant.
Author Response
Referee: 2
- Comment: Page 1, line 24 (Abstract): “Significantly” means that a statistically reduction occurred?
Response:
We are very sorry that we did not describe it clearly in the manuscript. “Significantly” means not a statistical reduction. Therefore,we have removed the “Significantly” according to your suggestions.
- Comment:Page 2, lie 40-41: I would say “Recent studies on probiotics proved that… “
Response:
Thanks for your suggestions. We have replaced it according to your suggestions.
- Comment: Page 2, line 51: I think that this sentence could be removed. It is clear that researchers are focusing on this topic.
Response:
Thanks for your suggestions. We have removed the “Experts and scholars have increasingly focused on postbiotics' health benefits in recent years” according to your suggestions.
- Comment: Page 2, line 53: The year of the study is missing. Please, add it.
Response:
Thanks for your suggestions. We have added the year of study on your comments.
- Comment: Page 2, lines 52-56: Specify which properties.
Response:
Thanks for your advice. We have revised the expression of properties according to your suggestions.
- Comment: Page 2, line 57: Replace “like” with “such as.”.
Response:
Thank you very much for pointing out that our wording was inaccurate. We have replaced it according to your suggestions.
- Comment: Page 2, line 60: The sentence could be removed.
Response:
Thanks for your suggestions. We have removed the “Briefly, postbiotics is an evolving concept that requires more research to be implemented.” according to your suggestions.
- Comment: Page 2, L 68: Add the year of the publication.
Response:
Thanks for your suggestions. We have added the year of study on your comments.
- Comment: Page 2, L72: “Significantly” means that a statistically reduction occurred? If it’s not a statistical reduction, please use adverbs such as “Relevantly” or “Importantly” or synonymous.
Response:
Thanks for your suggestions. “Significantly” means not a statistical reduction. Therefore,we have removed the “Significantly” of the new manuscript.
- Comment: Page 2, L75: Which previous studies? Can you add them as references?
Response:
Thanks for your suggestions. We have corresponded to specific references for each bacteria.
- Comment: Page 5, LL 171-172: “…more significant impact compared to ….” is not sufficiently clear. I suggest to specify how much the impact was more significant.
Response:
We are very sorry that we did not describe it clearly in the manuscript. As follows, we have redescribed the changes at 70 ℃.
“Notably, the application of heat treatment at 70 °C seemed to have some impact compared to ET-22 cells, resulting in increased surface roughnessd and white clustered particles on the basis of cell death.”
